# Daytime admission is associated with higher 1-month survival for pediatric out-of-hospital cardiac arrest: Analysis of a nationwide multicenter observational study in Japan

Mafumi Shinohara[1,2]*, Takashi Muguruma[1,2], Chiaki Toida[1,2], Masayasu Gakumazawa[1,2], Takeru Abe[1,2], Ichiro Takeuchi[1,2]

1 Advanced Critical Care and Emergency Center, Yokohama City University Medical Center, Yokohama, Kanagawa, Japan, 2 Department of Emergency Medicine, School of Medicine, Yokohama City University, Yokohama, Kanagawa, Japan

* s_mafumi@yokohama-cu.ac.jp

## Abstract

### Objective

Hospital characteristics, such as hospital type and admission time, have been reported to be associated with survival in adult out-of-hospital cardiac arrest (OHCA) patients. However, findings regarding the effects of hospital types on pediatric OHCA patients have been limited. The aim of this study was to analyze the relationship between the hospital characteristics and the outcomes of pediatric OHCA patients.

### Methods

This study was a retrospective secondary analysis of the Japanese Association for Acute Medicine-out-of-hospital cardiac arrest registry. The period of this study was from 1 June 2014 to 31 December 2015. We enrolled all pediatric patients (those 0–17 years of age) experiencing OHCA in this study. We enrolled all types of OHCA. The primary outcome of this study was 1-month survival after the onset of cardiac arrest.

### Results

We analyzed 310 pediatric patients (those 0–17 years of age) with OHCA. In survivors, the rate of witnessed arrest and daytime admission was significantly higher than nonsurvivors (56% vs. 28%, $p < 0.001$; 49% vs. 31%; $p = 0.03$, respectively). The multiple logistic regression model showed that daytime admission was related to 1-month survival (odds ratio, OR: 95% confidence interval, CI, 3.64: 1.23–10.80) ($p = 0.02$). OHCA of presumed cardiac etiology and witnessed OHCA were associated with higher 1-month survival. (OR: 95% CI, 3.92: 1.23–12.47, and 6.25: 1.98–19.74, respectively). Further analyses based on the time of admission showed that there were no significant differences in the proportions of patients with witnessed arrest and who received bystander cardiopulmonary resuscitation and emergency medical service response time by admission time.

**Data Availability Statement:** Data are available from the JAAM-OHCA registry committee (contact via http://www.jaamohca-web.com/) for

researchers who meet the criteria for access to confidential data. The data are owned and imposed by a third -party organization, the JAAM-OHCA registry committee.

**Funding:** The authors received no specific funding for this work.

**Competing interests:** The authors have declared that no competing interests exist.

## Conclusion

Pediatric OHCA patients who were admitted during the day had a higher 1-month survival rate after cardiac arrest than patients who were admitted at night.

## Introduction

The incidence of pediatric out-of-hospital cardiac arrest (OHCA) has been reported to be 8.0–10.6 per 100,000 persons per year [1, 2]. Although the survival rate of pediatric OHCA patients varies between age groups and the reasons for cardiac arrest [1], the rate of survival remains low (reported as 6.4%-8%) [1, 3, 4]. Pediatric OHCA is relatively rare compared to adult OHCA. OHCA patients aged under 20 years accounted for only 1.4% of all OHCA patients according to the Japanese annual ambulance report [5]. A previous study utilizing a national-based registry reported that the proportion of children less than 18 years old among total OHCA patients was 2.4% in Japan [6].

Admitting hospital characteristics, such as critical care medical centers, teaching hospitals, annual patient numbers, and admission time, have been associated with patient outcomes in adults with OHCA [7–9]. In particular, cardiac arrest centers were associated with favorable outcomes of survival after OHCA in areas of providing high-quality specialized treatment and postcardiac arrest care [10–13]. An association between the time of admission and survival after cardiac arrest has been discussed. Admission at night has been associated with a poor survival rate in both adult and pediatric OHCA patients, and prehospital factors and the number of staff members and the activity of medical staff might contribute to these results [14–19]. Meanwhile, there was no change in the rate of 1-month survival with a favorable outcome after adult OHCA, which might be explained by prehospital factors, including constant bystander cardiopulmonary resuscitation (CPR) rate and advanced life support (ALS) performance of emergency medical service (EMS) personnel during the day [20]. However, findings regarding the relationship between admitting hospital characteristics and pediatric OHCA patients have been limited. It is important to determine how admitting hospital characteristics are associated with pediatric OHCA survival. This study aimed to analyze the association between admitting hospital characteristics and outcomes in pediatric OHCA patients. The findings of this study could contribute to build an effective admission policy for pediatric OHCA patients in hospitals.

## Materials and methods

### Study design and setting

This study was a secondary data analysis of the Japanese Association for Acute Medicine-Out-of-Hospital Cardiac Arrest registry (JAAM-OHCA registry). The JAAM-OHCA registry is a nationwide registry of OHCA patients, that has been managed by the Japanese Association for Acute Medicine (JAAM). The methodology of the JAAM-OHCA registry has been described in a previous report [6]. The study period was from June 1, 2014 to December 31, 2015. All pediatric patients (aged 0–17 years) who experienced OHCA were enrolled in this study. We enrolled all types of OHCA (including cardiac and noncardiac etiology).

The primary outcome of this study was 1-month survival after the onset of cardiac arrest. The secondary outcome was a favorable neurological outcome 1-month after OHCA, defined as a pediatric cerebral performance category (PCPC) score [21, 22] 1 or 2.

## Emergency medical service system in Japan

Japan had a population of approximately 126.5 million in 2019. EMS is one of the sections of local government throughout the country. The emergency telephone number 119 is accessible anywhere in Japan and is free to call. When a 119 call is received in the dispatch center, the dispatcher orders the nearest available ambulance to head to the site. In each ambulance, at least three persons work as EMS staff. EMS staffs perform ALS according to the Japan Resuscitation Council Resuscitation Guidelines [23]. All of them can provide CPR according to the Japanese CPR guidelines. The trained EMS personnel called emergency life-saving technicians (ELSTs), are allowed to insert intravenous catheters, administer epinephrine, maintain the airways using devices including tracheal tubes, and provide defibrillation using semi-automated external defibrillators for OHCA patients. The basic policies of the EMS activity guidelines and the selection of the hospitals are decided beforehand according to a regional medical control association formed by emergency medical physicians, local government personnel and dispatch center personnel. There has been a small discrepancy in the basic policies of EMS activity among different areas in the nation. According to a previous report of a web-based survey of emergency medical supervisors of 767 fire defense headquarters in Japan, 82% answered that administration of epinephrine was limited for patients aged >8, and 12% of the supervisors answered that administration of epinephrine was limited for patients aged >15 [24]. In addition, some regions allow physicians to go out with their ambulances, although there is no nationwide regulation for this. The EMS staff select the accepting hospitals according to the patients' conditions while following the basic policies. In most cases the EMS team transports OHCA patients to the closest critical care facility.

## Pediatric emergency medical system in Japan

In Japan, the development of pediatric emergency medicine and critical care is patterned from Western countries. In a 2017 survey, the number of hospitals with a pediatric intensive care unit (PICU) and the number of PICU beds in Japan were only 8% and 20% of those in the US, respectively [25, 26]. In addition, hospitals with PICUs are not necessarily emergency medical centers. Of the 27 Japanese hospitals with PICU, only 11 (40%) had a pediatric emergency center [25]. Furthermore, local critical care facilities that did not have a PICU would accept both adult and child OHCA patients. In addition, many pediatric patients after cardiac arrest have been treated in an adult intensive care unit (ICU) or mixed ICU, but not in a PICU.

Regarding the working system of physicians in Japan, many Japanese hospitals have introduced a shift called "tochoku," meaning that the physician on the night shift continues from the morning to the morning of the next day [19, 27]. Traditionally, the working hours for the 24-h continuous night shift for emergency physicians in Japan starts at 9:00 a.m., and physicians work through the night until the next morning at 9:00 a.m. The day shift starts at 9:00 a.m. and continues until 5:00 p.m. In addition, the number of doctors is reduced at night. The number of laboratory technologists and radiologists, as well as nurses, is also reduced at night.

## Data collection and quality control

Prehospital data were obtained from the All-Japan Utstein Registry of the Fire and Disaster Management Agency (FDMA) of Japan, using the data form of the Utstein-style international guideline for reporting OHCA [28]. The JAAM-OHCA registry collected information on OHCA patients after hospital arrival (available from: http://www.jaamohca-web.com/download/). During the study period, the physician or medical staff in charge of the patient input anonymized data using a web form or fax. The data were logically checked by the system and confirmed by the JAAM-OHCA registry committee, composed of specialists in emergency

medicine and epidemiology. If the data form was incomplete, the committee returned it to the respective hospital to complete as much as possible. In-hospital data were systematically combined with prehospital Utstein-style data gathered by the FDMA using the following five key items: prefecture, emergency call time, age, sex, and PCPC one month after OHCA, which was evaluated by an attending physician.

The following factors related to admitting hospital characteristics for each patient were used: type of hospital (critical care medical center, and others); number of hospital beds and ICU beds; presence of a dedicated PICU; number of cardiac arrest patients transported in the previous year; number of pediatric cardiac arrest patients transported in the previous year; whether the emergency department (ED) had one or more pediatricians; whether the ED had one or more specialist physicians of emergency medicine; whether the ED had one or more specialist physicians of intensive care; number of physicians or nurses who attended to an OHCA case (daytime and nighttime duty); daytime admission (hospital arrival time between 9:00 a.m. and 4:59 p.m.); and whether one or more pediatricians participated during the resuscitation. Critical care medical centers (CCMCs) are tertiary medical facilities certified by the Japanese Ministry of Health, Labour and Welfare. To qualify as a CCMC, a hospital needs to have over 20 beds, an ICU, one or more emergency medical specialists and accept critically ill patients 24 h a day. In this study, we defined *pediatricians* as specialists in general pediatrics, not necessarily trained in intensive care, and *intensive care specialists* as general intensivist not necessarily trained in pediatric intensive care. The institutes were categorized into high-volume institutes (upper 50%) and low-volume institutes (lower 50%) depending on the number of total OHCA patients admitted in the previous year. In the same way, the institutes were categorized into pediatric high-volume institutes (upper 50%) and pediatric low-volume institutes (lower 50%) depending on the number of pediatric OHCA patients admitted in the previous year. The following items related to the patients' characteristics were considered: age, sex, cause of cardiac arrest (cardiac origin OHCA or other causes), whether OHCA was witnessed, bystander CPR, and initial rhythm and implementation of an automated external defibrillator (AED) by a layperson. Cardiac origin OHCA was caused by a cardiac disease (e.g., acute coronary syndrome, fatal arrhythmia, and congenital heart disease) and presumed cardiac cause (diagnosis made when no evidence of a noncardiac cause was found) clinically by a physician. The age groups were categorized as follows: infants (0–1 year), young children (2–7 years), older children (8–12 years), and teenagers (13-years). EMS response time was considered, as these would relate to the patients' outcome: time from call to EMS arrival and time from call to arrival at the hospital. The implementation of therapeutic hypothermia and extracorporeal membrane oxygenation (ECMO) were also considered as indicators of intensive care after cardiac arrest.

## Ethics approval

The JAAM-OHCA registry was approved by the Ethics Committee of Kyoto University [6]. The research period of this study was approved by the JAAM-OHCA registry committee from June 2014 to December 2015. This study was approved by the institutional review board of Yokohama City University Medical Center (D1502003).

## Statistical analysis

We compared the characteristics of survivors and nonsurvivors at 1 month after the onset of OHCA. Quantitative variables were expressed as medians [interquartile range, 25 to 75 percentiles] and compared using the Mann-Whitney U test. The comparisons for categorical variables were performed using the chi-square test or Fisher's exact test. Multivariate analysis was

performed using a multiple logistic regression model of primary and secondary outcomes to identify related factors. If we found significant factors of admitting hospital characteristics on survival from the multivariate analysis, we further analyzed other hospital or patient characteristics based on the significant variables. In this way, we were able to test whether the significant admitting hospital characteristics variables might be confounded by other characteristics. In addition, we compared the hospital and patient characteristics between patients who were admitted to the hospital with or without a PICU to identify whether the survival rate might be confounded by other characteristics. A *p* value of less than 0.05 was considered statistically significant. All statistical analyses were performed using STATA software (STATA/SE 13.0, Stata-Corp LLC, Texas, USA).

## Results

Between June 2014 and December 2015, 13,491 patients were registered in the JAAM-OHCA registry. A total of 319 children (0–17 years old) with OHCA were identified. We excluded nine patients whose in-hospital data were not recorded; therefore, we analyzed 310 patients with OHCA (Fig 1).

Of the 310 children with OHCA, 39 (12.6%) survived 1 month after OHCA, and 16 (5.2%) achieved favorable neurological outcomes at 1 month after OHCA. There were 200 (64.5%) males, and the median age [interquartile range] was 3 [0–14] years. The proportion of OHCA patients with a cardiac cause was 30.3%. The patients were treated in 55 hospitals. There were five hospitals with PICUs (9%). All receiving hospitals employed more than one emergency medicine specialist certified by the JAAM. Among the 55 receiving facilities, 49 had intensive care specialists, and 41 had pediatricians. The median numbers [interquartile range] of all cardiac arrest and pediatric cardiac arrest patients transported in the previous year were 150 [100–250] and 5 [2–6], respectively, for each accepting hospital. High-volume institutes received 69% of the patients. The number of patients treated in hospitals with a PICU was 31. Pediatricians were involved in treatment in 108 (35%) cases.

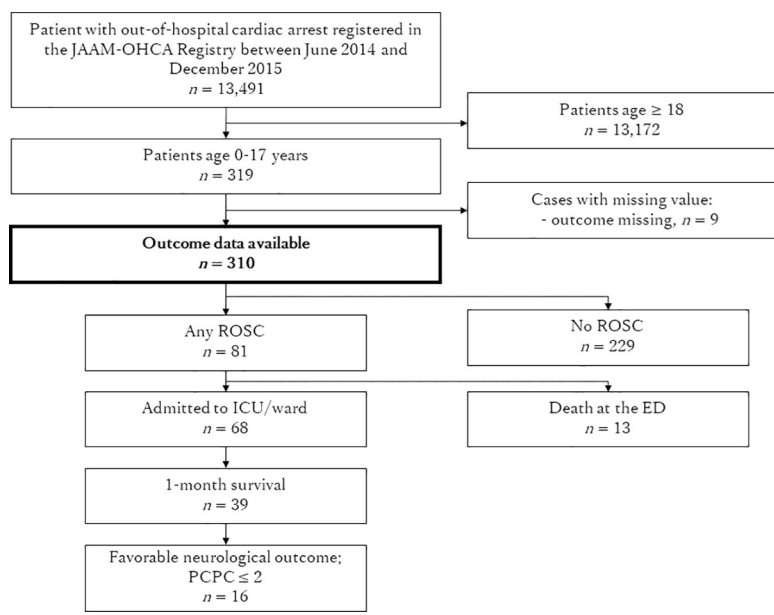

**Fig 1. Patients selection flow diagram.**

**Table 1. Characteristics of patients according to 1-month survival.**

| characteristics | survivors (n = 39) | | non-survivors (n = 271) | | *p* value |
|---|---|---|---|---|---|
| | frequency (%) | | frequency (%) | | |
| age group | | | | | 0.18 |
| infants (0–1 year) | 17 | (44%) | 122 | (45%) | |
| young children (2–7 years) | 2 | (5%) | 44 | (16%) | |
| older children (8–12 years) | 6 | (15%) | 23 | (9%) | |
| teenagers (13–17 years) | 14 | (36%) | 82 | (30%) | |
| sex male | 27 | (69%) | 173 | (64%) | 0.51 |
| cardiac origin OHCA | 17 | (44%) | 77 | (28%) | 0.05 |
| witnessed OHCA | 22 | (56%) | 75 | (28%) | 0.00 |
| admission to critical care medical center | 35 | (90%) | 256 | (95%) | 0.25 |
| admission to hospitals with a PICU | 1 | (3%) | 30 | (11%) | 0.10 |
| admission to high-volume institute | 27 | (69%) | 181 | (67%) | 0.76 |
| admission to pediatric high-volume institute | 24 | (62%) | 191 | (71%) | 0.26 |
| the emergency department had one or more specialist physicians of intensive care | 37 | (95%) | 254 | (94%) | 0.78 |
| the emergency department had one or more pediatricians | 25 | (64%) | 205 | (76%) | 0.12 |
| one or more pediatrician participated during the resuscitation | 14 | (36%) | 94 | (35%) | 0.32 |
| number of doctors, night | | | | | 0.67 |
| 1 | 4 | (10%) | 23 | (9%) | |
| 2 | 9 | (23%) | 81 | (30%) | |
| ≥3 | 26 | (67%) | 167 | (62%) | |
| number of nurses, night | | | | | 0.09 |
| 1 | 9 | (23%) | 111 | (4%) | |
| 2 | 19 | (49%) | 107 | (40%) | |
| ≥3 | 11 | (28%) | 53 | (20%) | |
| daytime admission (09:00 a.m.-04:59 p.m.) | 19 | (49%) | 85 | (31%) | 0.03 |

A comparison of patients who did and did not survive for 1 month is shown in Table 1. The rate of daytime admission was significantly higher in survivors (49% vs. 31%, *p* = 0.03). The rate of witnessed arrest was higher in survivors than in nonsurvivors (56% vs. 28%, *p* < 0.001). Other factors, such as critical care medical center or not, having a PICU, number of hospital beds, number of ICU beds, number of cardiac arrest patients transported in the previous year, whether the department had one or more pediatricians, whether one or more pediatricians participated during resuscitation and the number of physicians or nurses were not associated with 1-month survival.

The multiple logistic regression model showed that cardiac origin OHCA ((odds ratio, OR 3.92: 95% confidence interval, CI 1.23–12.74) (*p* = 0.02), witnessed arrest (OR 6.25: 95% CI 1.98–19.74) (*p* = 0.002), and daytime admission (OR: 95% CI, 3.64: 1.23–10.80) (*p* = 0.02) were associated with higher 1-month survival (Table 2).

Factors associated with favorable neurological outcomes 1 month after OHCA were analyzed. Results revealed that there were no significant variables associated with favorable neurological outcomes 1 month after OHCA.

Among the children with OHCA, 19/104 (18%) survived at 1 month after a daytime OHCA compared to 20/206 (10%) after nighttime arrest. The results suggested that the time of admission was related to the outcome of pediatric OHCA. We also compared the two groups, based on the time of admission. There were no significant group differences, exept for the proportion except for proportion of layperson AED applications (S1 Table), proportion of patients with

**Table 2. Factors associated with 1-month survival.**

| variables | odds ratio | 95% confidence interval | p value |
|---|---|---|---|
| age group | | | |
| infants (0–1 year) | 0.77 | (0.20–2.94) | 0.70 |
| young children (2–7 years) | 0.23 | (0.03–2.22) | 0.21 |
| older children (8–12 years) | 1.31 | (0.17–9.81) | 0.80 |
| teenagers (13–17 years) | reference | | |
| admission to critical care medical center | 0.54 | (0.04–7.96) | 0.65 |
| admission to pediatric high-volume institute | 0.68 | (0.20–2.25) | 0.53 |
| the emergency department had one or more pediatricians | 0.48 | (0.13–1.76) | 0.27 |
| one or more pediatrician participated during the resuscitation | 0.75 | (0.21–2.68) | 0.66 |
| admission to hospitals with a PICU | - | - | - |
| cardiac origin OHCA | 3.92 | (1.23–12.47) | 0.02 |
| witnessed OHCA | 6.25 | (1.98–19.74) | 0.002 |
| OHCA with bystander CPR | 3.18 | (0.99–10.20) | 0.05 |
| number of doctors, night | 0.66 | (0.25–1.71) | 0.39 |
| number of nurses, night | 1.29 | (0.53–3.15) | 0.57 |
| daytime admission (9:00 a.m.-04:59 p.m.) | 3.64 | (1.23–10.80) | 0.02 |

For covariate 'admission to hospitals with a PICU', no odds ratio was calculated because there was only one survivor in patients who admitted to hospitals with PICU one-month after cardiac arrest.

witnessed OHCA, and proportion who received bystander CPR according to admission time. Additionally, there were no significant differences in the rations of cardiac origin OHCA and EMS response time.

The number of patients treated in hospitals with a PICU was 31. Among them, only one patient survived 1 month after cardiac arrest. While there was no significant difference, the 1-month survival of patients treated in hospitals without a PICU was 11%. We also compared patient characteristics between the two groups, based on whether the accepting hospital had a PICU or not. In patients treated in the hospital with a PICU, a significantly higher rate of cardiac origin OHCA, admission to a pediatric high-volume institute with the presence of pediatricians, and treatment by pediatricians were observed (S2 Table).

## Discussion

In this study, we revealed that daytime admissions were associated with higher 1-month survival in pediatric OHCA patients (adjusted OR 3.64: 95% CI 1.23–10.8). Cardiac origin cardiac arrest and witnessed arrest were also associated with higher 1-month survival (adjusted OR: 95% CI, 3.92: 1.23–12.47, and 6.25: 1.98–19.74, respectively) as reported in a previous study [4]. There were no other factors associated with 1-month survival, including the type of hospital, number of patients, and number of attending physicians or specialties.

### Hospital type

Admitting hospital characteristics, such as critical care medical center, teaching hospitals and annual patient numbers have been reported to be associated with patient outcomes in adults with OHCA [7–9]. It was recently reported that the survival rate of nontraumatic OHCA was higher in the pediatric EDs than in general EDs, while the survival rate was not affected in non-transferred patients [2]. However, we did not find an association between the admitting hospital characteristics and outcomes of pediatric OHCA patients.

In this study, only 31 patients were treated in hospitals with a PICU. This small number of cardiac arrests in pediatric patients admitted to hospitals with PICUs may be related to the medical system in Japan. The number of PICUs is smaller in Japan than in Europe and the United States [25, 26, 29]. Because of the small number of PICU beds, the centralization of critically ill or injured children was insufficient in Japan [30]. Most pediatric OHCA patients were transported to the nearest emergency hospital without a PICU and treated. The median [interquartile] number of pediatric OHCA admissions to each hospital was 5 [2–6] per year. In a previous study, hospitals with higher experience of invasive mechanical ventilation showed lower mortality for pediatric OHCA [31]. This small number of pediatric cardiac arrests may make maintaining preparedness at each hospital for pediatric cardiac arrest and assuring high-quality pediatric postcardiac arrest care difficult.

## Time of admission and survival after cardiac arrest

The association between the time of admission and survival after cardiac arrest has been discussed. There aresome reports that admission at night has been associated with a poor survival rate in both adult and pediatric OHCA patients [14–19]. Previous studies have suggested some possible explanations for lower survival rates among adults with OHCAs at nighttime than during daytime. First, OHCA that occurs at night is witnessed less often than OHCA that occurs during the day, and patients who experience OHCA at night have a lower implementation rate of bystander CPR [15–17]. Second, the location where OHCA occurs differs depending on the time of day. Night OHCA events are likely to occur at home, and the implementation rate of AED is low [15, 17]. Third, reports show that EMS response times are longer at night than during the day [14, 15, 17]. Finally, the number of staff members and the activity of the staff may decrease during nighttime [15]. However, a recent study from Austria reported no change in the rate of 30-days survival with favorable outcomes after OHCA, and prehospital factors, including constant bystander CPR rate and ALS performance of EMS personnel were the presumed reason [20]. In pediatric cardiac arrest, Kitamura et al. reported that the 1-month survival rate of pediatric bystander-witnessed OHCA was lower during nighttime than during daytime [19]. It is important to identify the reason for the decreased survival rate after pediatric cardiac arrest at night. Studies of pediatric in-hospital cardiac arrest have shown that the survival rate for arrests was higher during the daytime than at nighttime [32, 33]. Esangbedo et al. reported that there was no difference in CPR quality depending on time of day [32]. Staffing and postcardiac arrest care have been mentioned as possible reasons [32, 33]. Kitamura et al. also suggested that the time of the day was related to postcardiac arrest care [19].

In our study, there were no significant differences in the rates of witnessed arrest and bystander CPR according to nighttime and daytime arrests. There was also no difference in the EMS response time. The rate of layperson AED application was high in daytime OHCA. However, there was no change in the rate of cardiac origin OHCA and the rate of patients whose initial rhythm was ventricular fibrillation or pulseless ventricular tachycardia (S1 Table). The difference in the level of intensive care between day and night, shortage of staff members, and decrease in staff performance at night are possible reasons for the difference in survival rates over time. However, it remains that prehospital confounders may be more markedly associated with outcome than in-hospital confounders.

Our findings can contribute to the development of an effective admission policy for pediatric OHCA patients in a hospital or a local area. Traditional emergency physicians in Japan who were on the night shift started their 24-h continuous shift at 9:00 a.m. and worked through the night until the next morning at 9:00 a.m. [19, 27]. The day shift starts at 9:00 a.m.

and continues until 5:00 p.m. In addition, the number of laboratory technologists and radiologists, as well as nurses, is reduced at night. This system may adversely affect care at night. Some hospitals have implemented 8-h or 12-h shift work to relieve the heavy burden of emergency physicians. A previous report showed that emergency medicine physicians worked less effectively when working on night shifts than when working on day shifts [15]. Improvements in hospital admission systems, including increasing the number of staff during the night and promoting early examination and treatment, may improve the outcome of pediatric OHCA patients. Further study is needed to determine the relationship between working hours and physician performance.

## Limitations

Our study had several limitations. First, although the data were obtained from a national-based database, there were few pediatric OHCA patients; and because of the small sample size, the power may be insufficient. In addition, the JAAM-OHCA registry did not cover all OHCA patients, and selection bias may have occurred. Second, there was no information on patients' baseline status or baseline PCPC data from the registry. It is possible that some patients with lower baseline PCPC scores were discharged with an identical PCPC but were counted as having a worse outcome. Third, the cause of OHCA (cardiac origin OHCA or not) was examined, and cardiac origin OHCAs were diagnosed clinically by the physician in charge of the clinical scene. However, OHCA caused by cardiac disease (acute coronary syndrome, fatal arrhythmia, and congenital heart disease) and presumed cardiac cause (diagnosis made when no evidence of a noncardiac cause was found) were included. Fourth, post-arrest care, such as respiratory and cardiac support and electrolyte and blood glucose management, were not examined in detail. While the quality of intensive care is important for survival after cardiac arrest [34], post-arrest care was not conducted uniformly across institutions. In addition, we did not have information on which patients were transferred to the PICU after receiving initial treatment. Furthermore, patients who were admitted at 4:59 p.m. (and earlier), could be treated "at nighttime". Patients who were admitted at early in the morning were treated "at daytime". This could have influenced our results. Fifth, there was no detailed information on whether senior ELSTs, senior hospital-based physicians, or hospital-based nurses were less available at nighttime in Japan, which might subsequently influence patient outcomes. Finally, this study was an observational study; therefore, causation could not be inferred. A relatively large study is needed in the future.

## Conclusions

Pediatric OHCA patients admitted during the day had a higher 1-month survival rate after cardiac arrest than patients admitted during the night. Improvements in hospital admission systems, including increasing the number of staff during the night and promoting early examination and treatment, might contribute to improve the survival of pediatric OHCA patients.

## Supporting information

**S1 Table. Characteristics of patients according to admission time.**
(DOCX)

**S2 Table. Characteristics of patients treated in the hospital with and without a PICU.**
(DOCX)

## Author Contributions

**Conceptualization:** Mafumi Shinohara, Takashi Muguruma.

**Data curation:** Mafumi Shinohara.

**Formal analysis:** Mafumi Shinohara, Takeru Abe.

**Investigation:** Mafumi Shinohara, Chiaki Toida, Masayasu Gakumazawa.

**Methodology:** Mafumi Shinohara.

**Project administration:** Takashi Muguruma, Ichiro Takeuchi.

**Supervision:** Ichiro Takeuchi.

**Validation:** Mafumi Shinohara.

**Visualization:** Mafumi Shinohara.

**Writing – original draft:** Mafumi Shinohara, Takeru Abe.

**Writing – review & editing:** Mafumi Shinohara, Takashi Muguruma, Chiaki Toida, Masayasu Gakumazawa, Takeru Abe, Ichiro Takeuchi.

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
