## [Decision Letter · Decision Letter 0]

14 Oct 2020

PONE-D-20-26211

Daytime admission is associated with higher survival for pediatric out-of-hospital cardiac arrest: Analysis of a nationwide multicenter observational study in Japan

PLOS ONE

Dear Dr. Shinohara,

Thank you for submitting your manuscript to PLOS ONE. After careful consideration, we feel that it has merit but does not fully meet PLOS ONE’s publication criteria as it currently stands. Therefore, we invite you to submit a revised version of the manuscript that addresses the points raised during the review process.

The reviewers point to several issues that need to be addressed. In particular, Reviewer #1 points to the need to include the location of the cardiac arrest as a potential confounder and suggests modifying the regression to include the interaction between time of day and admission hospital type.

We look forward to receiving your revised manuscript.

Kind regards,

Richard Bruce Mink

Academic Editor

PLOS ONE

Journal Requirements:

Reviewers' comments:

Reviewer's Responses to Questions

**Comments to the Author**

1. Is the manuscript technically sound, and do the data support the conclusions?

Reviewer #1: No

Reviewer #2: Yes

2. Has the statistical analysis been performed appropriately and rigorously? 

Reviewer #1: No

Reviewer #2: Yes

3. Have the authors made all data underlying the findings in their manuscript fully available?

Reviewer #1: Yes

Reviewer #2: Yes

4. Is the manuscript presented in an intelligible fashion and written in standard English?

Reviewer #1: Yes

Reviewer #2: Yes

5. Review Comments to the Author

Reviewer #1: The main objective of this study is unclear. It is likely that the authors are trying to investigate the association of type or grade of hospital with outcome of pediatric OHCA during daytime and nighttime. However, too many results are shown. Admission time should not be included in admission hospital characteristics. Analyses for interaction between time of day and admission hospital type is necessary and should be included in multivariable regression analysis. The authors should compare the outcomes of OHCA between type of admission hospital during daytime and nighttime.

Location of cardiac arrest, an important prehospital confounder for outcome of pediatric OHCA, particularly for schoolchildren cases is not included in analyses. Schoolchildren stay at school during school hours on schooldays.

The authors should logically re-analyze the data with consideration of interaction between hospital grade and time of day.

Minor

Page 4, Line 54 “the relationship between the admitting hospital characteristics and the outcomes of pediatric OHCA patients” should be read as “the association of admitting hospital characteristics with outcomes of pediatric OHCA patients”.

Page 4, Line 54 “the relationship between the admitting hospital characteristics and the outcomes of pediatric OHCA patients” should be read as “the association of admitting hospital characteristics with outcomes of pediatric OHCA patients”.

It is likely that prehospital confounders are more potently associated with outcome than in-hospital confounders. Please emphasize this in Discussion.

Reviewer #2: I congratulate the authors for their efforts in enroling a high number of patients in this study, as well as to push forward in emergency medicine research. I read your manuscript with great interest and came upon a few points that in my opinion should be cleared before further considering your work.

1) Title: Please state "1-month survival" or "30-days survival" instead of just "survival".

2) Abstract: Objective: I would rephrase this to: "Hospital characteristics such as hospital type [...]". Also, I would try to overcome the repitition of "hospital characteristics" so often.

3) Abstract: Methods: Please also state the type of OHCA (also traumatic?). Also include this information in your Conclusion.

4) Overall: Try to revise a few sentences so that word repetitions are not so obvious throughout the text.

5) Introduction: Lines 45-47: Try to integrate information on cardiac arrest centres and their impact here (for example 10.1016/j.resuscitation.2016.06.021).

6) Introduction: Lines 47-48: Try to describe this aspect from a more controversial side, as there is also evidence against the daytime impacting on outcomes (e.g.: 10.1016/j.resuscitation.2019.07.009).

7) Methods: Lines 61-62: I acknowledge that your detailed methodology has been previously reported, but please give a very short overview for a potential reader.

8) Methods: Line 63: Why not 0-18 years?

9) Methods: Line 67: I would rephrase "belongs".

10) Methods: Line 67: "was" accessible?

11) Methods: If your EMS can perform full ALS, please explicitly state so and cite the respective pediatric ALS guidelines. Also state, if you have any physicians going out with your ambulances (as in physician-based systems, outcomes may look different).

12) Methods: "Pediatric emergency...": I would try to shorten this section a bit (for example I do not think it is necessary to give details about the first PICU).

13) Methods: Lines 93-94: And also the number of doctors, right? Plus, are there fewer specialists and more doctors in training on call at night or does the ratio mirror the daytime situation?

14) Data collection: Line 106: Who accessed the CPC? And how was it accessed if the patient had already been discharged?

15) Data collection: Lines 108-110: I think this belongs in the Methods section.

16) Data collection: Line 112: It would be interesting to know the definitions of "critical care medical center" and so on.

17) Data collection: Line 118: When a patient was admitted at 4:59pm (and also earlier), they were surely treated "at nighttime", so this could have influenced your results. Please add this to your Limitations.

18) Results: Line 168: Is only 9% of the hospitals having a PICU not extremely low? Where were the children treated after CA if not at a PICU? Were they immediately transferred?

19) Results: Lines 175-180: I think you should also state here that there was a sign. difference in witnessed CA.

20) Results: Same as comment 19 should be applied in the odds ratios: There are several sign. associations apart from only daytime (in table 2), please also state those in the text.

21) Discussion: Lines 229-231: But in your table, you show other factors that are sign.?

22) Discussion: General: Please try to structure your discussion more in order to make it easier for a potential reader to follow (e.g. through subheadings and pooling of similar ideas); avoid "jumps of thought" and try to be as concise as possible.

23) Conclusion: It may not be clear to a reader what you mean by the sentence "Changing hospital [...]".

6. PLOS authors have the option to publish the peer review history of their article (what does this mean?). If published, this will include your full peer review and any attached files.

Reviewer #1: No

Reviewer #2: No

---

## [Author Response · Author response to Decision Letter 0]

25 Nov 2020

To the Editor:

We thank you for all of your reviews and thorough comments on our manuscript.

We numbered your comments for convenience.

Please find our responses to each comment below, in which we believe we have addressed all the concerns. 

In this letter, we use the number of lines from a clean copy of the revised manuscript. 

Reviewer Comments　

Reviewer #1

Major comment

Comment 1-1 The main objective of this study is unclear. It is likely that the authors are trying to investigate the association of type or grade of hospital with outcome of pediatric OHCA during daytime and nighttime. However, too many results are shown. Admission time should not be included in admission hospital characteristics. Analyses for interaction between time of day and admission hospital type is necessary and should be included in multivariable regression analysis. The authors should compare the outcomes of OHCA between type of admission hospital during daytime and nighttime.

Location of cardiac arrest, an important prehospital confounder for outcome of pediatric OHCA, particularly for schoolchildren cases is not included in analyses. Schoolchildren stay at school during school hours on schooldays.

The authors should logically re-analyze the data with consideration of interaction between hospital grade and time of day.

Response 1-1

Thank you for your comment. We re-performed the multiple logistic regression analysis with hospital type (critical care medical center or not) and an interaction term of hospital type and admission time. However, we found that no odds ratio was calculated for the interaction term because multi-collinearity occurred. In addition, there was no significant difference between time of admission and hospital type (p = 0.85, in a Chi-squared test). We re-analyzed the multiple logistic regression analysis, adding a hospital type (critical care medical center or not) as one of covariates, as well. In addition, there were no data of location of cardiac arrest in our dataset. Thus, we mentioned in the limitation that we were not able to analyze factor of the place of cardiac arrest. Those amended results from analysis was shown in Table 2, in our revised manuscript. 

Minor comment

Comment 1-2 

Page 4, Line 54 “the relationship between the admitting hospital characteristics and the outcomes of pediatric OHCA patients” should be read as “the association of admitting hospital characteristics with outcomes of pediatric OHCA patients”.

Response 1-2

Thank you for your comment. We changed the sentence, following your comment. 

Line 58- 60, p. 5

The aim of this study was to analyze the association of admitting hospital characteristics with outcomes of pediatric OHCA patients.

Comment 1-3 

It is likely that prehospital confounders are more potently associated with outcome than in-hospital confounders. Please emphasize this in Discussion.

Response 1-3

Thank you for your comment. As you pointed out, prehospital confounders are more potently associated with outcome than in-hospital confounders. Thus, we added the description of this in the discussion section. 

Line 306- 307, p. 19

However, it still remains that prehospital confounders may be more potently associated with outcome than in-hospital confounders.

Reviewer #2

I congratulate the authors for their efforts in enroling a high number of patients in this study, as well as to push forward in emergency medicine research. I read your manuscript with great interest and came upon a few points that in my opinion should be cleared before further considering your work.

Comment 2-1 

1) Title: Please state "1-month survival" or "30-days survival" instead of just "survival".

Response 2-1

Thank you for your comment. We changed the word “survival” into “1-month survival” in the title.

Line 1-2, p. 1

Daytime admission is associated with higher 1-month survival for pediatric out-of-hospital cardiac arrest: Analysis of a nationwide multicenter observational study in Japan

Comment 2-2

2) Abstract: Objective: I would rephrase this to: "Hospital characteristics such as hospital type [...]". Also, I would try to overcome the repetition of "hospital characteristics" so often.

Response 2-2

Thank you for your comment. We change the word “Admitting hospital characteristics such as hospital type […]” into “Hospital characteristics such as hospital type […]” in the abstract. In addition, we tried to avoid the repetition of “hospital characteristics”.

Line 17- 19, p. 2

Hospital characteristics, such as hospital type and admission time, have been reported to be associated with survival in adult out-of-hospital cardiac arrest (OHCA) patients. However, findings regarding the effects of hospital types on pediatric OHCA patients have been limited.

Comment 2-3 

3) Abstract: Methods: Please also state the type of OHCA (also traumatic?). Also include this information in your Conclusion.

Response 2-3 

Thank you for your comment. We added the statement that we enrolled all type of OHCA in the abstract and the methods section, as well.

Line 24- 25, p. 2

We enrolled all types of OHCA. 

Line 70- 71, p. 5

We enrolled all types of OHCA (including cardiac and non-cardiac etiology).

Comment 2-4

4) Overall: Try to revise a few sentences so that word repetitions are not so obvious throughout the text.

Response 2-4

Thank you for your comment. We tried to avoid word repetitions throughout the abstract.

Abstract

Objective: Hospital characteristics, such as hospital type and admission time, have been reported to be associated with survival in adult out-of-hospital cardiac arrest (OHCA) patients. However, findings regarding the effects of hospital types on pediatric OHCA patients have been limited. The aim of this study was to analyze the relationship between the hospital characteristics and the outcomes of pediatric OHCA patients.

Methods: This study was a retrospective secondary analysis of the Japanese Association for Acute Medicine-out-of-hospital cardiac arrest registry. Data were collected by using a data sheet from the participating facilities. The period of this study was from 1 June 2014 to 31 December 2015. We enrolled all pediatric patients (those 0 - 17 years of age) experiencing OHCA in this study. We enrolled all type of OHCA (including cardiac and non-cardiac etiology). The primary outcome of this study was 1-month survival after the onset of cardiac arrest.

Results: We analyzed 310 pediatric patients (those 0 - 17 years of age) with OHCA. In survivors, the rate of witnessed arrest and daytime admission was significantly higher than nonsurvivors (56% vs. 28%, p < 0.001: 49% vs. 31%; p = 0.03, respectively). The multiple logistic regression model showed that daytime admission was related to 1-month survival (odds ratio, OR: 95% confidence interval, CI, 3.64: 1.23 - 10.80) (p = 0.02). Cardiac origin cardiac arrest and witnessed arrest were also associated with higher 1-month survival (OR: 95% CI, 3.92; 1.23 - 12.47, and 6.25; 1.98 - 19.74, respectively). Further analyses based on the time of admission showed that there were no significant differences in the proportions of patients with witnessed OHCA and who received bystander cardiopulmonary resuscitation and emergency medical service response time by admission time.

Conclusion: Pediatric OHCA patients who were admitted during the day had a higher 1-month survival rate after cardiac arrest than patients who were admitted at night.

Comment 2-5

5) Introduction: Lines 45-47: Try to integrate information on cardiac arrest centres and their impact here (for example 10.1016/j.resuscitation.2016.06.021).

Response 2-5 

Thank you for your comment. We added the description of impact of cardiac centers for survival after OHCA. 

Line 48- 49, p. 4

In particular, cardiac arrest centers were associated with favorable outcome of survival after OHCA in the area providing high quality specialized treatment and post-cardiac arrest care [10-13]. 

Comment 2-6

6) Introduction: Lines 47-48: Try to describe this aspect from a more controversial side, as there is also evidence against the daytime impacting on outcomes (e.g.: 10.1016/j.resuscitation.2019.07.009).

Response 2-6

Thank you for your comment. We added the description of the controversial side of admission time on outcome after OHCA.

Line 53- 56 p. 4

On the other hand, there was no change in rate of 30-days survival with a favorable outcome after adult OHCA, and this might be explained by pre-hospital factors, including constant bystander cardiopulmonary resuscitation (CPR) rate and advanced life support (ALS) performance of emergency medical service (EMS) personnel in day [20].

Comment 2-7

7) Methods: Lines 61-62: I acknowledge that your detailed methodology has been previously reported, but please give a very short overview for a potential reader.

Response 2-7 

Thank you for your comment. We added a sentence that showed overview of JAAM-OHCA registry. 

Line 66- 67, p. 5

The JAAM-OHCA registry is a nationwide registry of OHCA patients, which has been managed by the Japanese Association for Acute Medicine (JAAM).

Comment 2-8

8) Methods: Line 63: Why not 0-18 years?

Response 2-8

Thank you for your comment. Regarding the age range used, we followed nationwide analyses of OHCA especially an original report of JAAM-OHCA registry’s profile, such as Kitamura T, et al. Acute Med Surg 2018; 25: 249-258. Thus, we set the age of children was 0-17 years. 

Comment 2-9

9) Methods: Line 67: I would rephrase “belongs”.

Response 2-9

Thank you for your comment. As you pointed out, we rephrased the word and revised the sentence. 

Line 77, p. 5- Line 78, p. 6

EMS is one of the sections of a local government throughout the country.

Comment 2-10

10) Methods: Line 67: "was" accessible?

Response 2-10 

Thank you for your comment. The correct expression is the present form. We changed “was” to “is”.

Line78- 79, p. 6

The emergency telephone number 119 is accessible anywhere in Japan for free.

Comment 2-11

11) Methods: If your EMS can perform full ALS, please explicitly state so and cite the respective pediatric ALS guidelines. Also state, if you have any physicians going out with your ambulances (as in physician-based systems, outcomes may look different).

Response 2-11

Thank you for your comment. EMS staffs perform advanced life support (ALS) according to the Japan Resuscitation Council Resuscitation Guidelines. Thus, we added the description in our revised manuscript. In addition, we mentioned that there has been a small discrepancy in basic policies of EMS activity among different areas in the nation. In addition, there is no nationwide system that any physicians go to the site with ambulance in Japan. We added the description in our revised manuscript.

Line 80- 81, p. 6

EMS staffs perform ALS according to the Japan Resuscitation Council Resuscitation Guidelines [23]. 

Line 85- 93, p. 6

The basic policies of EMS activity guidelines and the selection of the hospitals are decided beforehand according to a regional medical control association which was constructed by emergency medical physicians, local government personnel and dispatch center personnel. There has been a small discrepancy in basic policies of EMS activity among different areas in the nation. According to a previous report of a web-based survey for the emergency medical supervisors of 767 fire defense head-quarters in Japan, 82% of them answered that administration of epinephrine was limited for patients aged over eight, and 12% of the supervisor answered that administration of epinephrine was limited for patients aged over15 [24]. In addition, some region might allow physicians going out with your ambulances, but there is no nationwide regulation on it.

Comment 2-12

12) Methods: "Pediatric emergency...": I would try to shorten this section a bit (for example I do not think it is necessary to give details about the first PICU).

Response 2-12 

Thank you for your comment. As you suggested, we deleted some sentences to shorten this section.

Line 98-105, p. 7

In Japan, the development of pediatric emergency medicine and critical care is patterned from western countries. In a 2017 survey, the number of hospitals with a pediatric intensive care unit (PICU) and the number of PICU beds in Japan were only 8% and 20% those of in the US, respectively [25, 26]. In addition, hospitals with PICU do not necessarily obtain emergency medical centers. Of the 27 Japanese hospitals with PICU, pediatric emergency center was only 11 (40%) [25]. Furthermore, the local critical care facilities that often do not have PICU would accept OHCA patients in both adults and children. In addition, many of pediatric patients after cardiac arrest have been treated in adult intensive care unit (ICU) or mixed ICU, but not in PICU.

Comment 2-13

13) Methods: Lines 93-94: And also the number of doctors, right? Plus, are there fewer specialists and more doctors in training on call at night or does the ratio mirror the daytime situation?

Response 2-13

Thank you for your comment. We added the description about the number of doctors at night. We could not find a research investigating the rate difference between specialist doctors and trainee doctors in the time of day. Thus, we added this point to the materials and methods section and the limitation section.

Line 110- 111, p. 7

In addition, the number of doctors is reduced at night. The number of laboratory technologists and radiologists, as well as nurses, is also reduced at night.

Line 336- 338, p. 21

Fifth, there was no detailed information on whether senior ELSTs, senior hospital-based physicians, or hospital-based nurses were less available at nighttime in Japan, which might subsequently influence on the patients’ outcome.

Comment 2-14

14) Data collection: Line 106: Who accessed the CPC? And how was it accessed if the patient had already been discharged?

Response 2-14 

Thank you for your comment. An attending physician evaluated the outcome of the patients. Thus, we added the description in our revised manuscript.

Line 121- 124, p. 8

In-hospital data were systematically combined with prehospital Utstein-style data gathered by the FDMA using the following five key items: prefecture, emergency call time, age, gender, and PCPC one month after OHCA ,which was evaluated by an attending physician.

Comment 2-15

15) Data collection: Lines 108-110: I think this belongs in the Methods section.

Response 2-15

Thank you for your comment. As you suggested, we amended the sentence in our revised manuscript. 

Line 72- 74, p. 5

The primary outcome of this study was 1-month survival after the onset of cardiac arrest. The secondary outcome was a favorable neurological outcome 1-month after OHCA, defined as a pediatric cerebral performance category (PCPC) score [21,22] 1 or 2.

Comment 2-16

16) Data collection: Line 112: It would be interesting to know the definitions of “critical care medical center” and so on.

Response 2-16

Thank you for your comment. CCMCs are the tertiary medical facilities that certified by the Japanese Ministry of Health, Labour and Welfare. To be qualified as a CCMC, a hospital needs to have over 20 beds, intensive care unit, one or more emergency medical specialist and accept critically ill patients 24 hours a day. We also added the definitions of critical care medical center and the other hospitals in our revised manuscript. 

Line 133- 136, p. 8

Critical care medical centers (CCMCs) are the tertiary medical facilities that certified by the Japanese Ministry of Health, Labour and Welfare. In order to be qualified as a CCMC, a hospital needs to obtain over 20 beds, an intensive care unit, one or more emergency medical specialist and accept critically ill patients 24 hours a day.

Comment 2-17

17) Data collection: Line 118: When a patient was admitted at 4:59pm (and also earlier), they were surely treated "at nighttime", so this could have influenced your results. Please add this to your Limitations.

Response 2-17

Thank you for your comment. We added the description in the limitation section. 

Line 334-336, p. 21

Furthermore, patients who were admitted at 4:59pm (and also earlier) could be treated at nighttime. Patients who were admitted at early in the morning were treated at daytime, as well. They could have influenced our results.

Comment 2-18

18) Results: Line 168: Is only 9% of the hospitals having a PICU not extremely low? Where were the children treated after CA if not at a PICU? Were they immediately transferred?

Response 2-18

Thank you for your comment. In Japan, there were very small number of hospitals having a PICU. Many of pediatric patients after cardiac arrest have been treated in adult ICU or mixed ICU, not PICU. We added the explanation in the section of pediatric emergency medical system in Japan in our revised manuscript. In addition, we did not have data which patients were transferred to PICU after receiving initial treatment. Thus, we described it in the limitation section. 

Line 103- 105, p. 7

In addition, many of pediatric patients after cardiac arrest have been treated in adult ICU or mixed ICU, but not in PICU.

Line 332- 334, p. 21

In addition, we did not have information which patients were transferred to PICU after receiving initial treatment.

Comment 2-19

19) Results: Lines 175-180: I think you should also state here that there was a sign. Difference in witnessed CA.

Response 2-19

Thank you for your comment. As you pointed out, witnessed arrest is associated with higher one-month survival in pediatric OHCA. Thus, we added the description regarding this point in the result section and in the abstract.

Line 27- 29, p. 2

In survivors, the rate of witnessed arrest and daytime admission was significantly higher than nonsurvivors (56% vs. 28%, p < 0.001: 49% vs. 31%; p = 0.03, respectively).

Line 192- 193, p. 11

The rate of witnessed arrest was higher in survivors than nonsurvivors (56% vs. 28%, p < 0.001).

Comment 2-20

20) Results: Same as comment 19 should be applied in the odds ratios: There are several sign. associations apart from only daytime (in table 2), please also state those in the text.

Response 2-20

Thank you for your comment. We added the description of signs that associated with 1-month survival in the result section and in the abstract.

Line 29- 32, p. 2

The multiple logistic regression model showed that daytime admission was related to 1-month survival (odds ratio, OR: 95% confidence interval, CI, 3.64: 1.23 - 10.80) (p = 0.02). Cardiac origin cardiac arrest and witnessed arrest were also associated with higher 1-month survival (OR: 95% CI, 3.92: 1.23 - 12.47, and 6.25: 1.98 - 19.74, respectively).

Line 201- 204, p. 12

The multiple logistic regression model showed that cardiac origin OHCA ((odds ratio, OR 3.92: 95% confidence interval, CI 1.23 - 12.74) (p = 0.02), witnessed arrest (OR 6.25: 95% CI 1.98 - 19.74) (p = 0.002), and daytime admission (OR: 95% CI, 3.64: 1.23 - 10.80) (p = 0.02) were associated with higher 1-month survival (Table 2).

Comment 2-21

21) Discussion: Lines 229-231: But in your table, you show other factors that are sign.?

Response 2-21

Thank you for your comment. We stated the other factors that associated with higher one-month survival in pediatric OHCA. 

Line 248, p. 16- Line 250, p. 17

Cardiac origin cardiac arrest and witnessed arrest were also associated with higher 1-month survival (adjusted OR: 95% CI were 3.92: 1.23 - 12.47, and 6.25: 1.98 - 19.74, respectively) as previous study [4]. 

Comment 2-22

22) Discussion: General: Please try to structure your discussion more in order to make it easier for a potential reader to follow (e.g. through subheadings and pooling of similar ideas); avoid “jumps of thought” and try to be as concise as possible.

Response 2-22

Thank you for your comment. As you suggested, we re-structured our discussion with subheadings, “Hospital type”, “Time of admission and survival after cardiac arrest” and “Limitation” in our revised manuscript.

Comment 2-23

23) Conclusion: It may not be clear to a reader what you mean by the sentence "Changing hospital [...]".

Response 2-23

Thank you for your comment. As you pointed out, it was not clear initially. Thus, we amended the sentence in our revised manuscript.

Line 343-345, p. 21

 Improvements in hospital admission systems, including increasing the staff during the night and promoting early examination and treatment, might contribute to improve the survival of pediatric OHCA patients.

---

## [Decision Letter · Decision Letter 1]

31 Dec 2020

PONE-D-20-26211R1

Daytime admission is associated with higher 1-month survival for pediatric out-of-hospital cardiac arrest: Analysis of a nationwide multicenter observational study in Japan

PLOS ONE

Dear Dr. Shinohara,

Thank you for submitting your manuscript to PLOS ONE. After careful consideration, we feel that it has merit but does not fully meet PLOS ONE’s publication criteria as it currently stands. Therefore, we invite you to submit a revised version of the manuscript that addresses the points raised during the review process.

There are still a few minor issues that need to be addressed. In addition, as noted by Reviewer #2, please have the manuscript reviewed by an individual with expertise in written English.

We look forward to receiving your revised manuscript.

Kind regards,

Richard Bruce Mink

Academic Editor

PLOS ONE

Reviewers' comments:

Reviewer's Responses to Questions

**Comments to the Author**

1. If the authors have adequately addressed your comments raised in a previous round of review and you feel that this manuscript is now acceptable for publication, you may indicate that here to bypass the “Comments to the Author” section, enter your conflict of interest statement in the “Confidential to Editor” section, and submit your "Accept" recommendation.

Reviewer #1: All comments have been addressed

Reviewer #2: All comments have been addressed

2. Is the manuscript technically sound, and do the data support the conclusions?

Reviewer #1: Yes

Reviewer #2: Yes

3. Has the statistical analysis been performed appropriately and rigorously? 

Reviewer #1: Yes

Reviewer #2: Yes

4. Have the authors made all data underlying the findings in their manuscript fully available?

Reviewer #1: Yes

Reviewer #2: Yes

5. Is the manuscript presented in an intelligible fashion and written in standard English?

Reviewer #1: Yes

Reviewer #2: Yes

6. Review Comments to the Author

Reviewer #1: Thank you for your efforts. The manuscript has been improved. However. I still have the following concern about your manuscript.

MAJOR

The main findings of this study are

1. Daytime admission was related to 1-month survival (odds ratio, OR: 95% confidence interval, CI, 3.64: 1.23 - 10.80)

2. Cardiac origin cardiac arrest and witnessed arrest were also associated with higher 1-month

survival (OR: 95% CI, 3.92: 1.23 - 12.47, and 6.25: 1.98 - 19.74, respectively).

Thus, Tables and 2 are main tables. I recommend that Tables 3 and 4 are numbered as supplemental tables. Also, detailed explanation for supplemental tables are unnecessary.

MINOR

Abstract Results

“Cardiac origin cardiac arrest and witnessed arrest were also associated with

higher 1-month survival” should be read as “OHCA of presumed cardiac etiology and witnessed OHCA were associated with higher 1-month survival

Reviewer #2: Dear Authors,

Thank you for addressing all of my comments and suggestions, I feel that the manuscript has improved substantially.

However, I would suggest having another look at the English grammar and style of your manuscript, as there are still a few sentences and passages that read confusing (e.g., presence or absence of articles,... etc.). The best way to overcome this would be involving a native English speaker.

7. PLOS authors have the option to publish the peer review history of their article (what does this mean?). If published, this will include your full peer review and any attached files.

Reviewer #1: No

Reviewer #2: No

---

## [Author Response · Author response to Decision Letter 1]

25 Jan 2021

To the Editor:

We thank you for all of your reviews and thorough comments on our manuscript.

We numbered your comments for convenience.

Please find our responses to each comment below, in which we believe we have addressed all the concerns. 

In this letter, we use the number of lines from a clean copy of the revised manuscript. 

Reviewer Comments　

Reviewer #1: Thank you for your efforts. The manuscript has been improved. However. I still have the following concern about your manuscript.

Comment 1-1

MAJOR

The main findings of this study are

1. Daytime admission was related to 1-month survival (odds ratio, OR: 95% confidence interval, CI, 3.64: 1.23 - 10.80)

2. Cardiac origin cardiac arrest and witnessed arrest were also associated with higher 1-month

survival (OR: 95% CI, 3.92: 1.23 - 12.47, and 6.25: 1.98 - 19.74, respectively).

Thus, Tables 1 and 2 are main tables. I recommend that Tables 3 and 4 are numbered as supplemental tables. Also, detailed explanation for supplemental tables are unnecessary.

Response 1-1

Thank you for your comment. We numbered Tables 3 and 4 from the previous version as S1 and S2 Tables in the Supplementary Material and removed the detailed explanation from the Results and the Discussion section.

1) We deleted and amended below in the Results: 

Line 212, p. 13- Line 224, p. 14

Among the children with OHCA, 19/104 (18%) survived at 1 month after a daytime OHCA compared to 20/206 (10%) after nighttime arrest. The results suggested that the time of admission was related to the outcome of pediatric OHCA. We also compared the two groups, based on the time of admission. There were no significant group differences, exept for the proportion except for proportion of layperson AED applications (S1 Table), proportion of patients with witnessed OHCA, and proportion who received bystander CPR according to admission time. Additionally, there were no significant differences in the rations of cardiac origin OHCA and EMS response time. 

The number of patients treated in hospitals with a PICU was 31. Among them, only one patient survived 1 month after cardiac arrest. While there was no significant difference, the 1-month survival of patients treated in hospitals without a PICU was 11%. We also compared patient characteristics between the two groups, based on whether the accepting hospital had a PICU or not. In patients treated in the hospital with a PICU, a significantly higher rate of cardiac origin OHCA, admission to a pediatric high-volume institute with the presence of pediatricians, and treatment by pediatricians were observed (S2 Table).

2) We deleted below from the previous version in the Discussion:

Perhaps unexpectedly the nominal one-month survival rate was 3% among patients treated in hospitals with a PICU versus 11% in hospitals without a PICU, (p = 0.10). Although this was not a statistically significant difference, we further examined factors that might have been associated with survival rates in hospitals with PICU versus those without. There was no difference in age group or gender, and there was no difference in the rate of witnessed cardiac arrest. The EMS response time also did not differ. There was a difference in the rate of cardiac origin OHCA. Although detailed diagnosis was unknown, those cases might have a history of heart disease that had already been known. In addition, patients sent to a hospital with PICU may have had a higher rate of having some underlying diseases.

Comment 1-2

MINOR

Abstract Results

“Cardiac origin cardiac arrest and witnessed arrest were also associated with

higher 1-month survival” should be read as “OHCA of presumed cardiac etiology and witnessed OHCA were associated with higher 1-month survival.

Response 1-2

Thank you for your comment. We changed the sentence, following your comment. 

Line 31- 32, p. 2 

OHCA of presumed cardiac etiology and witnessed OHCA were associated with higher 1-month survival.

Reviewer #2: 

Comment 2-1

Dear Authors,

Thank you for addressing all of my comments and suggestions, I feel that the manuscript has improved substantially.

However, I would suggest having another look at the English grammar and style of your manuscript, as there are still a few sentences and passages that read confusing (e.g., presence or absence of articles,... etc.). The best way to overcome this would be involving a native English speaker.

Response 2-1

Thank you for your comment. We utilized an English editing service to improve our manuscript. We attached the editing certification for it.

---

## [Editor Report · Decision Letter 2]

28 Jan 2021

Daytime admission is associated with higher 1-month survival for pediatric out-of-hospital cardiac arrest: Analysis of a nationwide multicenter observational study in Japan

PONE-D-20-26211R2

Dear Dr. Shinohara,

We’re pleased to inform you that your manuscript has been judged scientifically suitable for publication and will be formally accepted for publication once it meets all outstanding technical requirements.

Kind regards,

Richard Bruce Mink

Academic Editor

PLOS ONE
---

## [Editor Report · Acceptance letter]

1 Feb 2021

PONE-D-20-26211R2 

Daytime admission is associated with higher 1-month survival for pediatric out-of-hospital cardiac arrest: Analysis of a nationwide multicenter observational study in Japan 

Dear Dr. Shinohara:

I'm pleased to inform you that your manuscript has been deemed suitable for publication in PLOS ONE. Congratulations! Your manuscript is now with our production department. 

Kind regards, 

on behalf of

Dr. Richard Bruce Mink 

Academic Editor

PLOS ONE